# Malnutrition and Gut Microbiota in Children

**DOI:** 10.3390/nu13082727

**Published:** 2021-08-08

**Authors:** Ishawu Iddrisu, Andrea Monteagudo-Mera, Carlos Poveda, Simone Pyle, Muhammad Shahzad, Simon Andrews, Gemma Emily Walton

**Affiliations:** 1Department of Food and Nutritional Sciences, University of Reading, Whiteknights, Reading RG6 6AX, UK; I.Iddrisu@pgr.reading.ac.uk (I.I.); a.monteagudo@reading.ac.uk (A.M.-M.); c.g.povedaturrado@reading.ac.uk (C.P.); 2Unilever R&D, Colworth Park, Sharnbrook, Bedfordshire MK44 1LQ, UK; Simone.Pyle@unilever.com; 3Institute of Basic Medical Sciences, Khyber Medical University, Peshawar 25100, Pakistan; shahzad.ibms@kmu.edu.pk; 4School of Biological Sciences, University of Reading, Whiteknights, Reading RG6 6AX, UK; s.c.andrews@reading.ac.uk

**Keywords:** severe acute malnutrition, gut microbiota, children, infants, breastfeeding, diet

## Abstract

Malnutrition continues to threaten the lives of millions across the world, with children being hardest hit. Although inadequate access to food and infectious disease are the primary causes of childhood malnutrition, the gut microbiota may also contribute. This review considers the evidence on the role of diet in modifying the gut microbiota, and how the microbiota impacts childhood malnutrition. It is widely understood that the gut microbiota of children is influenced by diet, which, in turn, can impact child nutritional status. Additionally, diarrhoea, a major contributor to malnutrition, is induced by pathogenic elements of the gut microbiota. Diarrhoea leads to malabsorption of essential nutrients and reduced energy availability resulting in weight loss, which can lead to malnutrition. Alterations in gut microbiota of severe acute malnourished (SAM) children include increased *Proteobacteria* and decreased *Bacteroides* levels. Additionally, the gut microbiota of SAM children exhibits lower relative diversity compared with healthy children. Thus, the data indicate a link between gut microbiota and malnutrition in children, suggesting that treatment of childhood malnutrition should include measures that support a healthy gut microbiota. This could be of particular relevance in sub-Saharan Africa and Asia where prevalence of malnutrition remains a major threat to the lives of millions.

## 1. Introduction

The prevalence of malnutrition in children under five years of age in developing nations remains alarming. The World Health Organization (WHO), United Nations Children’s Fund (UNICEF) and the World Bank Group reported that globally in 2017, 150.8 million (22.2%) children under five years of age were stunted, with wasting threatening the lives of 7.5% (50.5 million) of children [1]. Africa and Asia are the continents most affected by such malnutrition, bearing 39% and 55% (respectively) of global stunting prevalence in children under five. It is estimated that about half of all under-five-year-old childhood deaths are as a result of stunting (chronic malnutrition), culminating in about 3 million child deaths every year [1]. The most common and immediate causes of malnutrition/undernutrition in children are inadequate dietary intake and disease (such as diarrhoea), according to the UNICEF conceptual framework of the causes of malnutrition [2], resulting in deficient growth and development. Although inadequate access to sufficient nutrition is the main cause of malnutrition, gut microbiota have also been implicated in this condition [3,4]. Alterations in the gut microbiota, characterized by increases in the phylum Proteobacteria and decreases in *Bifidobacterium* and *Lactobacillus* species, are associated with episodes of diarrhoea [5,6,7], and diarrhoea is a major causative factor in malnutrition of children, especially those in low-income countries [8,9]. The causes of malnutrition, together with strategies to counteract malnutrition, are of significant public health interest, and manipulation of the gut microbiota provides a potential opportunity for alleviation of malnutrition [10].

The scope of this review therefore includes the role of diet in modifying the gut microbiota and how the gut microbiota impacts malnutrition in children. It also considers how the use of ready-to-use therapeutic foods (RUTF) in the treatment of severe acute malnutrition changes the gut microbiota of malnourished children.

## 2. The Human Gut Microbiome in Health and Disease

### 2.1. In Health

The human gastrointestinal tract harbours microbial populations consisting of 10^13^–10^14^ cells [11], but the density and composition differ markedly between compartments. In the stomach of healthy people, the bacterial load is relatively low at 10^2^ colony-forming units (CFUs)/mL of contents, but rises in the small intestine to 10^2^–10^4^ CFUs/mL [12]. However, the highest levels (10^12^ CFUs/mL) are found in the colon, where conditions (pH 5.5–7, slow transit time, and high nutrient availability) are highly conducive for bacterial growth [13]. The colonic bacteria are mainly anaerobic [12] and carry out a range of metabolic processes, some of which are considered of benefit to the host. For example, gut microbes ferment indigestible carbohydrates, generating short-chain fatty acids (SCFAs) for the host. These SCFAs have been reported to have several health benefits to the host, including the provision of energy for epithelial cells and lowering the pH of the intestinal lumen, thus restricting growth of some pathogens [14,15] and providing an anti-inflammatory effect to the host [16]. The gut microbiota has also been observed to play an important role in absorption, storage and expenditure of energy from the diet [17,18,19] as well as synthesis of vitamins K and B12 [20,21]. Energy intake and expenditure are predominantly coordinated by the brain. The gastrointestinal tract (GI), the first site of contact with ingested food, sends important signals (through neuroendocrine and neuroimmune systems) to the brain regarding the composition and size of incoming food [22]. Furthermore, the gut microbiota is involved in the production of metabolites, including SCFAs, secondary bile acids, tryptophan and even neurotransmitters; these play an important role in gut barrier function and also in gut–brain signalling [23,24]. The gut–brain axis is important for maintaining energy balance and controlling energy expenditure, and has also been observed to impact on cognitive function [25]. Such benefits have led to a growing interest in the use of prebiotics, probiotics and other dietary modifications to modulate the gut microbiota to improve nutrition and health [26].

The role of the gut microbiota of newborns in growth and development may be dependent on the acquired microbial composition [27]. Recent metagenomic analysis of the gut microbiota of infants suggests that in addition to the composition, the functional potential of the microbiota plays a significant role in the nutritional status of the infant. For instance, the development (transmitted from the mother) of bile acid and starch metabolism bacteria may impact nutrient absorption and therefore, affect growth and development of infants [28,29]. In breastfed infants, the gut microbiota is dominated by *Bifidobacterium longum* subsp. *infantis,* which is tailored towards the metabolism of human milk oligosaccharides (HMO) [30,31]. For infants fed on formula milk, infant formulae are commonly fortified with prebiotics so as to confer similar beneficial components as those found in breastmilk [32,33,34]. Prebiotics are “substrates that are selectively utilized by host microorganisms conferring health benefits to the host” [35]. The common prebiotics used in infant formula are short-chain galacto-oligosaccharides (GOS) and long-chain fructo-oligosaccharides (FOS). These prebiotics have been shown to beneficially alter the gut microbiota of infants by selectively enhancing the growth of *Bifidobacterium* and lowering the abundance of *Enterococcus* and *Escherichia coli* [36,37]. However, GOS and FOS are structurally different from HMO [38] and so do not entirely replicate the beneficial effects of HMOs on the infant gut microbiota, for example, on a species level [34,39,40]. Nevertheless, it has been noted that an infant diet supported by prebiotic formula results in fewer infections than a placebo formula does, and so non-HMO prebiotics appear to have a positive impact in infants [41].

Probiotics are defined as “live microorganisms that, when administered in adequate amounts, confer a health benefit on the host” [42]. It is now well established that probiotics can modulate the gut microbiota of the host in a beneficial fashion [43,44,45]. For instance, the addition of probiotics to infant formula has been known to confer numerous benefits to the infant, including the improvement of gut health and immunity, countering the growth of harmful bacteria (pathogens) in the gut and enhancing overall host immune and health status [43]. Moreover, the addition of probiotics including *Lactobacillus reuteri, Lactobacillus acidophilus, Bifidobacterium longum, Bacillus clausii* and *Bifidobacterium lactis* in infant formula has been shown to reduce risks associated with diarrhoea resulting from antibiotic use and the symptoms of colic [46,47,48]. In contrast, numerous studies have failed to find any effect of probiotics on diarrhoea outcomes in children [39,49,50]. Indeed, a recent Cochrane review based on large clinical trials with low risk of bias [51] reported little or no effect of probiotics in the reduction of diarrhoea. Such contradictory results are likely linked to the use of different probiotics and target populations. Given the contrasting evidence on the use of probiotics in the management of diarrhoea in children, there remains a need for targeted large-scale trials to follow on from the promising findings of specific probiotics that improve diarrhoea outcomes, especially in children.

### 2.2. In Disease

There is evidence linking the gut microbial community to a range of diseases and disorders, including inflammatory bowel disease (IBD), mood disorders, obesity, autism and psoriatic arthritis [52,53,54,55,56,57], though further research is needed to ascertain the causal link between the gut microbiota and such diseases. In some of these conditions, interventions that impact the gut microbial community have led to improvement of symptoms, further supporting a role for the microbiota [58,59,60]. An altered gut microbiota can be caused by environmental factors such as antibiotic use, diet and stress, as well as genetic factors. These changes can impair the ability of the gut microbiota to maintain good health and may allow the growth of potentially pathogenic bacteria (e.g., *Clostridioides difficile*), leading to production of metabolites that may cause a disease state in the host [61]. The relative abundance of bifidobacteria in faecal samples of normal-body-weight children was found to be higher than in overweight children of the same age bracket (7 years old). In contrast, *Staphylococcus aureus* levels were higher in the overweight children than in those with normal body weight [56]. In addition, a recent study reported an increased abundance of Firmicutes and a reduction in bifidobacteria in the gut microbiota of overweight and obese children [62]. Furthermore, it has been observed that children who became overweight/obese at age 10 years had significantly higher levels of *Bacteroides fragilis* in early infancy (3 to 6 weeks after birth) than those whose body weight remained normal [63]. These studies thus indicate that the microbial community of children may differ according to BMI, suggesting that the microbiota may have a part to play in weight gain, for example, through energy salvage. However, caution is required when considering causal links between the gut microbiota and obesity, as recent meta-analyses failed to find variations in the taxonomic microbial compositions of obese and lean adults, suggesting that microbiota differences observed in other studies could be related to factors such as diet [64,65,66]. Several other conditions, including eczema, asthma, inflammatory bowel disease (IBD) and type 1 diabetes, in infants and young children have been linked to differences in the gut microbiota. For instance, some studies show that infants with eczema have a significant reduction in relative abundance of *Bifidobacterium, Blautia, Coprococcus, Eubacterium* and *Propionibacterium* species [67] as well as a reduction in intestinal microbial population diversity [68,69]; although it is worth noting that in healthy infants, there is low microbiota diversity predominated by bifidobacteria. Moreover, an analysis of faecal microbiota of infants (1 to 11 months old) revealed that gut microbiota alteration promotes the disfunction of CD4^+^ T cells, which is linked to atopy in children [70]. Further, many studies have reported the gut microbiota to be altered in IBD [71,72,73,74], with an increase in Enterobacteriaceae being particularly prominent [75]. In addition, it has been found that children with prediabetes have higher levels of intestinal Bacteroidetes than healthy controls [76].

Thus, there is much evidence to suggest that the gut microbiota provides numerous health benefits to children, whilst an altered balance, as characterized by altered abundance of key species (e.g., pathogenic groups such as *Clostridioides difficile*), can be associated with negative health outcomes. However, one drawback of focusing on changes in the microbiota is that there is a difficulty in determining the clinical relevance of such changes, and so it remains unclear to what degree the gut microbiota contributes to malnutrition in children (see Section 4.3).

## 3. Diet and Gut Microbiota in the First 1000 Days of Life

### 3.1. Factors Affecting the Gut Microbiota of Infants

The first major microbial colonization of the human gut occurs at birth, with composition subsequently changing over time. A number of factors affect microbial colonization of the gut after birth, including gestational age, mode of delivery, maternal body mass index (BMI) and gut microbiota, infant feeding practices, antibiotic exposure, infant genetics, random exposure to microbes in the environment and the number of siblings; see Figure 1 [77,78,79,80,81]. Additionally, the type of delivery (vaginal or caesarean delivery) is an important factor that determines the early colonizers of the gut of newborns. Children who are born through normal vaginal birth (NVB) initially receive their gut microbiota from maternal vaginal and faecal sources, whilst those born through caesarean section are initially colonized by microbiota related to the mother’s skin [78,82,83,84]. In a recent longitudinal study, delayed microbiota colonization and significant taxonomic differences were observed for caesarean deliveries compared to vaginal deliveries in infants during their first year, although such differences were lessened 5 months after birth [85]. Moreover, a recent clinical study on the effect of mode of delivery on gut microbiota composition [86] showed an enrichment of *Bifidobacterium* spp. and a reduction in *Klebsiella* and *Enterococcus* spp. in the gut microbiota of vaginally delivered infants compared to infants born by caesarean section. This has also been shown in a more recent study [87] where infants born by caesarean section were found to be less likely to have *B. thetaiotaomicron* or *B. fragilis* in their gut microbiota. Additionally, a systematic review on the effect of mode of delivery on the gut microbiota of infants indicates a significant difference between vaginal- and caesarean-delivered children in the first 3 months, although the mode of delivery had less impact later in infancy (6–12 months) when complementary foods are usually introduced [88]. Furthermore, a longitudinal cohort study observed no differences in the gut microbiota between caesarean- and vaginal-delivered babies beyond 6 weeks of birth [28]. These studies indicate that the impact of delivery mode on the gut microbiota is lessened or lost as time progresses.

### 3.2. Impact of Breastfeeding on the Infant Gut Microbiota

Apart from being a source of energy and vital nutrients for infants, breastmilk also contains a diverse microbiome and oligosaccharides with prebiotic properties [81]. However, the human milk microbiome varies in response to maternal BMI, geographical location, duration of pregnancy, lactation and mode of delivery [77,78,79]. Many butyrate-producing bacteria such as *Roseburium* and *Eubacterium* have recently been demonstrated to grow on complex HMO [84]. In addition, *Bifidobacterium* and *Lactobacillus* spp. are present in breastmilk, and these are also found in maternal and neonatal faeces [82,83], which suggests their potential transmission from mother to child through breastfeeding [85]. These groups of bacteria have long been noted for their beneficial role in inhibiting pathogen colonization and in the production of vitamins, the modulation of the immune response and the maintenance of intestinal barrier function [88,89,90]. *B. longum* subsp. *infantis* and *Lactobacillus acidophilus* have been observed to be higher in abundance in the gut microbiota of breast-fed compared with formula-fed infants, whereas the abundance of *C. difficile, Enterobacter cloacae, Citrobacter* spp., *Granulicatella adiacens* and *Bilophila wadsworthia* is higher for formula-fed compared with breast-fed infants. Additionally, Ruminococcaceae, *Bacteroides* spp. and Lachnospiraceae of the Clostridia order are of raised abundance in the gut microbiota of infants on complementary feeding [91,92,93,94]. A recent clinical study observed an enrichment of *Bifidobacterium* and *Bacteroides* spp., and a lower abundance of *Streptococcus* and *Enterococcus* spp., in the gut microbiota of breast-fed compared with formula-fed infants [95]. In addition, as indicated above, breast milk has been observed to contain bifidobacteria and to promote a bifidogenic effect [83]. Thus, breastfeeding has a major impact on the composition of the infant gut microbiota, driving a predominance of bifidobacteria, during the first year of life [96]. It is also becoming evident that initial feeding practices and their resulting impact on the gut microbiota of infants have both short- and long-term effects on health in later childhood [41,56,97]. This was shown in a placebo-controlled, double-blinded study [98] where infants at risk of atopy were fed prebiotic-supplemented formula (containing 0.8 g/100 mL GOS/FOS) or a control formula (containing 0.8 g/100 mL maltodextrin) during the first 6 months after birth. They were then followed until age 5 years. The prebiotic-supplemented group had significantly lower incidence (five-year cumulative incidence) of allergic and atopic dermatitis manifestation compared with the control group [98]. This study emphasizes the long-term impact of early life feeding practice on the gut microbiota and health in children.

### 3.3. Effect of Diet on the Gut Microbiota of Infants and Young Children

There is an increase in infant gut microbiota diversity after the age of 6 months when complementary foods (with high protein and fibre) are introduced to the diet, indicating a transition to a more complex microbiota [92,99]. Thus, by 3 years of age, the gut microbiota assumes an adult-like composition [20,93,100], although other reports suggest that the gut microbiota does not become adult-like until the age of 5 years [101,102,103,104]. As noted above, the major driver of gut microbiota shift in infants is the introduction of complementary feeding during the first years [92]. For example, a randomised control trial showed that the introduction of complementary feeding (including cereal, fruit and meat) led to increased gut microbiota diversity [105]. In addition, it was previously shown that the introduction of complementary feeding in infants results in changes in the gut microbiota with increases in the genera *Bacteroides* and *Prevotella* [100,106]. Moreover, a comparative study of the gut microbiota of children (1 to 6 years old) in a rural area of Burkina Faso (BF) and an urban area of Italy revealed higher Actinobacteria and Bacteroidetes levels and lower Firmicutes and Proteobacteria numbers in the rural BF children relative to the urban Italian children. However, there were no differences in the microbiota between the two populations during the period of exclusive breastfeeding. Thus, the differences in the gut microbiota appeared to be associated with the differences in postweaning diets; the diets of the BF children were higher in dietary fibre and lower in fat and calories, and that of the European (EU) children was higher in fat, sugars and animal protein but lower in fibre [107]. The distinct microbiomes of the two groups of children likely impacts host metabolism and health; for example, the abundance of Actinobacteria and Bacteroidetes would allow maximisation of energy intake from dietary fibre and would proffer a protective effect on inflammation influences through the generation of SCFAs [107]; see Figure 2. Other studies showing that a postweaning diet impacts the infant gut microbiota include a recent study in Australian children (age 1 to 2 years), which showed the consumption of meat/fish and fruit (unprocessed foods) to be positively associated with Bacteroidetes (*Bacteroides thetaiotaomicron*) as well as *Ruminococcus*, *Lachnospiraceae* and *Bacteroides* species [108].

There is thus considerable evidence indicating that the composition of the gut microbiota of children adapts to the diet, and this would be expected to have an impact on the metabolic performance of the gut microbiota as well as on the health of the child [93]. This suggests that a deeper understanding of how diet affects the gut microbiota of infants and young children could lead to innovative approaches for improving health and may pave the way for new strategies for fighting malnutrition in children.

## 4. The Gut Microbiota and Malnutrition in Young Children

### 4.1. Malnutrition

Malnutrition is the condition where there is an imbalance in the nutrient/energy intake of an individual resulting from insufficient or excess consumption of nutrients. Malnutrition presents in several forms, including undernutrition (underweight, wasting, stunting, and micronutrient deficiency including vitamin and mineral deficiencies) and overnutrition (overweight and obesity) [109]. There are three main ways to assess the nutritional status of children. One involves anthropometric measurements, which include weight, height or length, mid-upper arm circumference and head circumference [109]. These are used to determine macronutrient malnutrition (related to carbohydrates, proteins and fats), especially in children. Other methods utilise clinical symptoms, such as presence or absence of bilateral oedema, to determine malnutrition. A further approach is the use of biochemical methods (such as haemoglobin, blood ferritin and retinol-binding protein levels), which are particularly useful in diagnosis of micronutrient malnutrition. Although clinical signs can indicate micronutrient malnutrition, biochemical tests are used to confirm or determine specific micronutrient deficiencies [110]. There are three main indices of malnutrition:(i).Weight-for-age (underweight) is, as the name implies, a measure of the child’s weight relative to their age compared to the WHO growth standard reference. Children whose weight-for-age falls more than two or three standard deviations (SD) below the average are classified as being underweight or severely underweight, respectively [109].(ii).Height-for-age (stunting) is a measure of the height of the child relative to the age compared to the WHO’s growth standards reference for children of the same age and sex. Children whose height-for-age Z-score is two or three SD below the average are classified as moderately or severely stunted, respectively [109]. This is a measure of long-term nutritional deficiency.(iii).Weight-for-height (wasting) measures acute malnutrition in children. It is the weight of the child compared to that of standard reference weights of healthy children of the same sex and height [109]. This is an indicator of recent or recurrent infections or diarrhoea because diarrhoea causes rapid weight loss, especially in children [111].

### 4.2. Causes of Malnutrition

Malnutrition is caused by different factors and, according to UNICEF 1990, these can be grouped into three main categories: (i) immediate causes, (ii) underlying causes and (iii) basic causes. The immediate causes are inadequate dietary intake and illness (e.g., diarrhoea). The underlying causes of malnutrition are household food insecurity, poor maternal and childcare practices, poor access to health and poor sanitation. These are factors that directly cause inadequate dietary intake and diseases among children. The basic causes, also known as root causes, are the resources, political, cultural and economic structures in the area [2].

### 4.3. Gut Microbiota in Malnutrition

Until a few decades ago, inadequate food intake and disease were the two recognised immediate causes of malnutrition, but enteric infections such as diarrhoea have since been reported to be a factor in a significant proportion of global malnutrition cases in children [112], suggesting a link between diarrhoea and malnutrition as caused by changes in the gut microbiota (see Figure 2).

Diarrhoea illness can lead to malnutrition through reduced nutrient absorption and mucosal damage, as well as through nutrient depletion associated with each episode of diarrhoea [112]. Diarrhoea in children leads to reduced weight and height gains, especially in children with recurrent diarrhoea. Several studies have shown that there are changes in the gut microbiota during and after diarrhoea illnesses [113,114,115]. For instance, consistent elevation of *Fusobacterium mortiferum*, *Escherichia coli* and oral microorganisms in the faecal microbiome of children (1–6 years old) suffering from diarrhoea have been reported [113]. Moreover, in a one-year cohort study of diarrhoea aetiologies in children [116], *Fusobacterium* from the phylum Fusobacteria was overrepresented in stool samples of diarrhoeal children compared to children who were recovering. In addition, a high relative abundance of *Bacteroides* in diarrhoea compared to recovery stools was observed [116]. Moreover, data from the Global Enterics Multicenter Study (GEMS) in West and East Africa, and Southeast Asia suggested that moderate-to-severe diarrhoea in children leads to reduced bacterial diversity and altered microbiota composition [117]. It was also reported that diarrhoea in children under 2 years of age from resource-poor areas may lead to decreased height (on average, 8 cm shortfall) and a decrease of 10 intelligence quotient (IQ) points by the age of 7 to 9 years [118]. As such, malnutrition-induced decreases in IQ are likely to impact quality of life in many ways, which could be countered by controlling diarrhoea and other factors associated with childhood malnutrition.

The above observations suggest that children suffering from diarrhoea during early life may suffer impairments in gut microbiota development that could result in persistent diarrhoea, leading to stunting, reduced IQ and growth faltering. Therefore, it is possible that pre- and probiotic interventions may help to support the microbiota to combat diarrhoea and offer some benefits against malnutrition [119]. The use of prebiotics may be of particular benefit to the management of SAM where children receive antibiotics as part of the treatment process. Prebiotics have been shown to reduce the adverse effects of antibiotics on the gut microbiota [120]; however, further research is needed with regards to their impact in the treatment of malnutrition.

It has been reported that certain subclinical alterations in the gut microbiome can lead to stunting even in the absence of obvious infections such as diarrhoea [121,122,123]. For instance, poor sanitary conditions whereby there is chronic exposure to environmental pathogens resulting in subclinical alteration in the gut microbiota structure and function have been proposed to cause stunting [124,125]. This has been termed environmental enteric dysfunction (EED). EED is defined as an acquired subclinical disorder of the small intestine, marked by villous atrophy and crypt hyperplasia. EED often affects children in resource-poor countries [121] and may contribute to stunting. In Gambian infants, faecal neopterin, which has been linked with cell-mediated inflammation and a marker of EED, was negatively associated with weight and height gain [126]. Consistently, a pilot study conducted in eight different countries observed a significant association between faecal neopterin, myeloperoxidase and alpha-1-antitrypsin, and a decrease in length-for-age z-scores (an indicator of stunting in children) [127].

Recent advances in understanding of the gut microbiome have revealed that the gut microbiota may be a contributor to undernutrition (see [10] for a recent review). In a longitudinal study [3], the gut microbiota of Malawian twins was associated with kwashiorkor (a form of severe acute malnutrition). Further, the gut microbiota of children who became malnourished in their first 3 years was poorly matured (in terms of microbial diversity) and different to that of healthy children (see Table 1). When applied to gnotobiotic mice, the combination of Malawian diet and kwashiorkor microbiota caused severe weight loss (about 35% weight loss from the initial body weight), suggesting a causal relationship between the gut microbiota and malnutrition. In addition, several studies in children from sub-Saharan Africa, India and Bangladesh have reported decreased bacterial diversity in faecal samples of children suffering malnutrition with increases in Proteobacteria, Bacteroidetes and anaerobic Firmicutes [128,129,130,131,132]. Consistent with these findings, it was also observed that Proteobacteria dominate the gut microbiota of hospitalized malnourished children in Uganda [133], as can be seen in Table 1. Moreover, in low-income countries, the prevalence of stunting increases in children of ages between 6 and 59 months [134,135,136], which corresponds to the period that directly follows the introduction of complementary feeding when the gut microbiota undergoes a major change in composition. The introduction of complementary feeding also increases the risk of diarrhoea, a risk factor for stunting, especially in resource-poor communities where high diarrhoea incidence is combined with inadequate nutrient intake [135,136]. Indeed, there is evidence to suggest that differences in the gut microbiota of infants in developing countries are associated with diarrhoea, which increases the risk of malnutrition (wasting and stunting) and mortality in affected children [123,137,138,139,140]. The first step in the dysbiosis of the gut microbiota during malnutrition in children is thought to be the depletion in bifidobacteria, which is followed by the colonization by potential microbial pathogens (*Streptococcus* spp., *Fusobacterium mortiferum* and *Escherichia coli*), causing diarrhoea and malabsorption of essential nutrients [116,128]. A similar depletion of beneficial species, and increase in potentially harmful species, was reported more recently, showing that malnourished children suffer a consistent reduction of species from the families Bacteroidaceae, Ruminococceae, Eubacteriaceae and Lachnospiroceae with an enrichment of the species *Staphylococcus aureus*, *Escherichia coli* and *Enterococcus faecalis* [141]. Moreover, it was shown that malnutrition in children is linked to reduction in the species *Bifidobacterium longum* and *Lactobacillus mucosae* [142].

**Table 1 nutrients-13-02727-t001:** Summary of the main findings of studies on infant and young child gut microbiota alteration during malnutrition.

Author(s), Subjects and Country	Study Design	Method of Analysis	Dietary Intervention	Main Findings
Smith, Yatsunenko [3] 317 Monozygotic and dizygotic twin pairs (<3 years) Côte d’Ivoire	Longitudinal study (follow-up until 36 months old) and transplantation of GM from twin pairs discordant for kwashiorkor to GF mice	16S rRNA large-scale sequencing and shotgun pyrosequencing	SAM children received ready-to-use therapeutic food (RUTF) and MAM received soy–peanut ready-to-use supplementary food. GF mice were given Malawian diet followed by RUTF	RUTF resulted in the maturation of the SAM microbiota with increases in *Bifidobacterium*, *Ruminococcus*, *Faecalibacterium prausnitzii* and *Lactobacillus* spp. and reduction in the members of the *Bacteriodetes*. *Bilophila wadsworthia* (*Proteobacteria, Desulfovibrio*) and *Clostridioides innocuum* were associated with kwashiorkor (SAM) microbiota
Subramanian, Huq [106] 64 SAM children 6–20 months oldBangladesh	Randomised intervention study	16S rRNA amplicon sequencing	RUTF compared to locally produced therapeutic food (Khichuri-Halwa) made from rice and lentils	Microbiota diversity inversely and significantly associated with Weight-for-Height Z-score (WHZ). SAM children had less diverse microbiota
Gough, Stephens [143] 18 Malawian twin pairs cohort and 11 Bangladeshi cohort Malawi and Bangladesh	Analysis of secondary data from case-control longitudinal cohort study of twins	Malawian twin cohorts: whole-genome shotgun sequencing datasets and Bangladeshi cohorts: relative abundance (OTUs)	Malawian cohort: not available Bangladeshi cohort: RUTF	Stunting severity (SAM) was associated with reduced gut microbiota diversity. In both cohorts, linear future growth is associated with relative abundance of *Acidaminococcus* spp. Bangladeshi cohort: largest decreases in relative abundance of *Dorea, Lactobacillus, Blautia, Olsenella* and other unclassified genera in the *Enterococcaceae* and *Coriobacteriaceae* in cases vs. controls whilst in the Malawian cohort, there were significant decreases observed in *Eubacterium, Prevotella, Blautia* and *Bacteroides* in cases vs. controls.
Ghosh, Sen Gupta [128] 20 children ages 5 to 60 months old India	Cross-sectional study	16S DNA large-scale sequencing	No specific dietary intervention	Borderline and severely malnourished children’s gut microbiota is characterised by increased abundance of *Shigella*, *Enterobacter*, *Veillonella*, *Escherichia*, *Streptococcus* and *Faecalibacterium. Roseburia* and *Butyrivibrio* characterised the gut microbiota of healthy children.
Monira, Nakamura [129] 7 healthy and 7 SAM children (ages 2 to 3 years old) Bangladesh	Cross-sectional study	16S rRNA amplicon sequencing (V5–V6 regions)	Both groups given normal Bangladeshi diet	SAM: the phyla Bacteroidetes and Proteobacteria accounted for 18% and 46%, respectively, of bacteria population Healthy: 44% and 5% Bacteroidetes and Proteobacteria, respectively
Kristensen, Wiese [133] 87 hospitalised SAM children (ages 6–24 months) with and without oedema Uganda	Cross-sectional study	Denaturing gradient gel electrophoresis (DGGE) and 16S rRNA amplicon sequencing (V3–V4 regions	F75 and F100 during the stabilization phase and RUTF during rehabilitation phase	Proteobacteria dominated the gut microbiota of hospitalized oedematous and non-oedematous SAM children and alpha diversity higher in only oedematous SAM children
Million, Tidjani Alou [141] 86 SAM from Senegal (*n* = 32) and Niger (*n* = 54) Also included a meta-analysis of 184 children from 5 studies (107 SAM; 77 healthy) Malawi, Bangladesh, Senegal, Niger and India	Case-control study	16S rRNA amplicon sequencing (V3–V4 regions)	Dietary intervention in Senegal was an energetic milk made of milk, oil and sugar. In Niger, children did not receive dietary treatment before recruitment	In SAM cases, there was depletion of several obligate anaerobes, including *Bacteroidaceae, Ruminococcaceae, Eubacteriaceae*, *Lachnospiraceae*, *Erysipelotrichaceae, Coriobacteriaceae* and *Eggerthella*. Additionally, enrichment in potential pathogenic bacteria species, including *Staphylococcus aureus, Enterococcus faecalis* and *Escherichia coli,* observed in SAM children
Dinh, Ramadass [142] 10 low-birth-weight children with persistent stunting (SAM), compared to 10 normal-birth-weight children (healthy with no stunting controls), followed till 2 years of age India	Longitudinal case control	16S rRNA amplicon sequencing (V4 region)	All children were breastfed for varying periods, complementary foods consisted of mainly rice, lentils and vegetables	There was higher relative abundance of the phylum Bacteroidetes in the stunted (SAM) children compared with control children at 12 months of age. The stunted (SAM) children’s microbiota was enriched with *Desulfovibrio* genus and Campylobacterales order, whereas the controls had enrichment in potentially probiotic species *Bifidobacterium longum* and *Lactobacillus mucosae*.
Tidjani, Million [132] 10 SAM children with oedema and 5 healthy children (2 to 38 months old) Senegal and Niger	Cross-sectional study	Culturomics using 18 different conditions. 16S rRNA sequencing targeting V3–V4 regions	No dietary intervention	Significantly, more *Streptococcus gallolyticus,* Proteobacteria and Fusobacteria as well as decrease in diversity of gut microbial population in the SAM children observed. A total of 45 species of bacteria identified in healthy children but missing in SAM microbiota, with 12 species found to have probiotics potential (*Bifidobacterium adolescentis, Anaerostipes caccae, Bacillus licheniformis, Bacillus subtilis, Intestinimonas butyriciproducens, Bacteroides salyersiae, Lactobacillus vaccinostercus, Terrisporobacter glycolicus, Alistipes indistinctus, Lactobacillus parabuchneri* and *Lactobacillus perolens*)
Raman, Gehrig [144] 36 members of the Bangladesh birth cohort	Sparse random forests (RF)-derived model	V4-16*S* rDNA amplicons	No intervention	A total of 15 bacteria taxa were identified to be associated with different degrees of undernutrition (SAM and moderate acute malnutrition—MAM). Termed “ecogroup”, the 15 bacteria taxa identified are *S. thermophilus F. prausnitzii* (514940 and 851865), *Bifidobacterium, E. coli*, *Dialister*, *B. longum*, *S. gallolyticus*, *Prevotella*, *P. copri* (840914 and 588929), *Clostridiales*, *E. rectale*, *L. ruminis* and *E. faecalis.*
Chen, Mostafa [145] 126 slum-dwelling children, Bangladesh	Randomised controlled trial	V4-16S rRNA gene sequencing	Microbiota-directed complementary food-2 compared with ready-to-use supplementary Food	MDCF-2 produced better outcomes with regards to wasting and stunting than RUSF did. MDCF-2 was associated with *Faecalibacterium prausnitzii*, *Dorea formicigenerans*, *Ruminococcus gnavus* and a member of *Clostridiales* previously described as weight discriminatory taxa
Gehrig, Venkatesh [146] Mice and Piglets model and undernourished children, Bangladesh	double-blind, randomized, four-group, parallel assignment interventional trial	V4-16S rRNA gene sequencing	MDCFs compared with RUSF	*Enterococcus avium*, *Escherichia fergusonii, Bifidobacterium pseudocatenulatum, Streptococcus constellatus* and *Streptococcus pasteurianus* were characteristic of malnourished microbiota. MDCF-2 was shown to increase the relative abundance of *F. prausnitzii*, *Clostridiales* sp., and decrease the abundance of *B. longum*. MDCF-2 shifted the plasma proteome of MAM toward a healthy one and improved growth parameters in MAM

It remains unclear whether the observed microbial differences contribute to malnutrition or whether the malnutrition is the cause of these changes, although experiments in mice demonstrated that a gut microbiota with relatively high differential abundance of *Bilophila wadsworthia* and *Clostridioides innocuum* combined with a low-nutrient diet enhanced malnutrition [3], which provides evidence of a role for the microbiota in mediating malnutrition. It must be noted that there is some variation in the gut microbiota alterations reported in different populations. However, those groups of bacteria consistently reported to decrease during malnutrition include *Bifidobacterium* spp., *Lactobacillus* spp., *Ruminococcus* spp. and *Faecalibacterium prausnitzii*, whilst those consistently reported to increase include Bacteroidetes, *Clostridioides innocuum, Streptococcus* spp. and *Escherichia* spp. (see Figure 1 for summary of bacteria consistently shifting during malnutrition).

In summary, the available evidence indicates a clear relationship between the gut microbiota and malnutrition. Thus, it is now recognized that malnutrition in children is linked to altered development and maturity of the gut microbiota, and as such, this is a functional target that should be explored in the fight against malnutrition [147].

### 4.4. Treatment of Malnourished Children with RUTF/MDCF Impacts the Gut Microbiota

Although the approach used to treat malnutrition in children varies from country to country, ready-to-use therapeutic foods (RUTF, a peanut-paste mixed with milk and vegetable oil, and fortified with vitamins and minerals that is used to treat children with severe acute malnutrition) has revolutionized the treatment of SAM during both in- and outpatient care [148,149]. In line with the recommendations of WHO inpatient treatment of SAM, a milk-based formula (F75) is used for the stabilization phase of treatment before energy/micronutrient-dense RUTF treatment is initiated. For community-based treatment of moderate acute malnutrition (MAM), ready-to-use supplementary foods (RUSF) are used. Over the years, compared to homemade formulae RUTF has been observed to be a more effective treatment for children with SAM. This has been reported in a recent randomized controlled trial in malnourished Ghanaian children, where standard RUTF treatment resulted in significant recovery (in terms of weight and mid-upper arm circumference) compared to locally formulated RUTF [150]. It was further shown that RUTF treatment of kwashiorkor-suffering Malawian twins from birth until 3 years resulted in the maturation of the microbiota with increases in *Bifidobacterium*, *Ruminococcus*, *Faecalibacterium prausnitzii* and *Lactobacillus* spp. and reduction in the members of the Bacteroidetes [3]. These changes are predicted to protect against enteropathogens by contributing to the production of bacteriocins and supporting the stimulation of the immune system. Such changes were only partially maintained once RUTF was withdrawn. The findings are consistent with results from a Bangladeshi study that examined the microbiota of malnourished and healthy children in an attempt to define gut microbiota maturation over the first 3 years of life [106]. Treatment of malnourished children with either RUTF or a locally made formula (made from rice, wheat flour, lentils, green leafy vegetables and soybean oil) resulted in a more diverse and mature microbiota, which regressed when treatment was stopped. In addition, a time series metagenomic analysis of faecal samples collected from the above Malawian study [3] showed lower gut microbial diversity in the SAM children compared with their healthy siblings. In contrast to these studies, a clinical study observed that RUTF treatment of kwashiorkor-suffering malnourished children from birth until 3 years of age did not lead to significant changes in gut microbiota diversity and composition [130].

A recent study comparing the effects of two “microbiota-directed complementary foods” (MDCF) and RUTF formulae in Bangladeshi children recovering from SAM confirmed the underdeveloped (immature) nature of the gut microbiome in SAM and post-SAM, MAM children (in comparison with an age-matched control group) [146]. This study further showed that an “optimized” MDCF formula (MDCF-2; containing flour made from chickpea, soyabean, green banana and peanut) can improve a range of growth and other health indicators in MAM children towards the status of healthy controls; this effect was associated with changed abundance of three bacteria most strongly associated with gut microbiota maturation (i.e., increased abundance of *F. prausnitzii* and *Clostridioides* spp., and decreased *B. longum*). A more recent randomized control trial further compared the effect of MDCF-2 and RUSF on growth and health recovery in MAM Bangladeshi children. This research confirmed that MDCF-2 treatment supports growth recovery of MAM children, which was linked to corresponding changes in the faecal microbiota abundance of 23 weight-discriminatory taxa [145]. The above reports suggest that MDCF treatment can support the gut microbiota of malnourished children. Thus, dietary targets that impact the microbiota are worthy of study in order to identify factors that counter malnutrition through microbiota-driven effects. Table 1 summarises the studies (as discussed in this review) on alterations of the gut microbiota of infants and young children during malnutrition.

An important implication of the above studies is that the beneficial changes of dietary interventions (such as RUTF) on the gut microbiota in the treatment of SAM are not sustained once treatment is withdrawn, and further work is required to confirm any beneficial long-term impact of such interventions relating to the gut microbiota of SAM children. However, the recent findings that MDCFs have the potential to “repair” the gut microbiota impairments of childhood malnutrition, leading to improved growth and development, raise the promise of improved future intervention strategies.

In summary, there is evidence, albeit limited [128,129,130,131], that directly links negative changes in the gut microbiota to malnutrition in children, which in turn raises the possibility of a relationship that could be manipulated to better manage health. Additionally, MDCF treatment of malnutrition presents a great opportunity, and clinical studies in different populations are warranted to confirm the effectiveness of this strategy. Further, although RUTF treatment results in good nutritional outcomes (anthropometric indices), there is limited data on the effect of RUTF treatment on the gut microbiota of malnourished children, indicating that more research is needed in this area.

## 5. Conclusions

In conclusion, diet is a strong modifier of the gut microbiota, especially in young children whose microbiota is subject to rapid changes. There is strong evidence linking the gut microbiota composition to malnutrition in children. Therefore, future interventions that support a healthy microbiota in malnourished children should be investigated as a tool to help to combat malnourishment. Any advancement in combating malnutrition would be particularly relevant to children in sub-Saharan Africa and Asia where the prevalence of malnutrition continues to threaten the lives of millions of children.

### Future Direction

Targeting the gut microbiota, such as through the use of prebiotics, as part of the treatment of childhood malnutrition could present an effective, affordable and sustainable strategy in the treatment and/or prevention of SAM. Fortifying the microbiota in this way could enhance energy salvaging and may protect against diarrhoea. However, clinical evidence that can be translated into clinical practice is lacking. Therefore, studies on the impact of prebiotics on the gut microbiota and its influence on nutrition outcomes in children suffering from malnutrition are needed.

## Figures and Tables

**Figure 1 nutrients-13-02727-f001:**
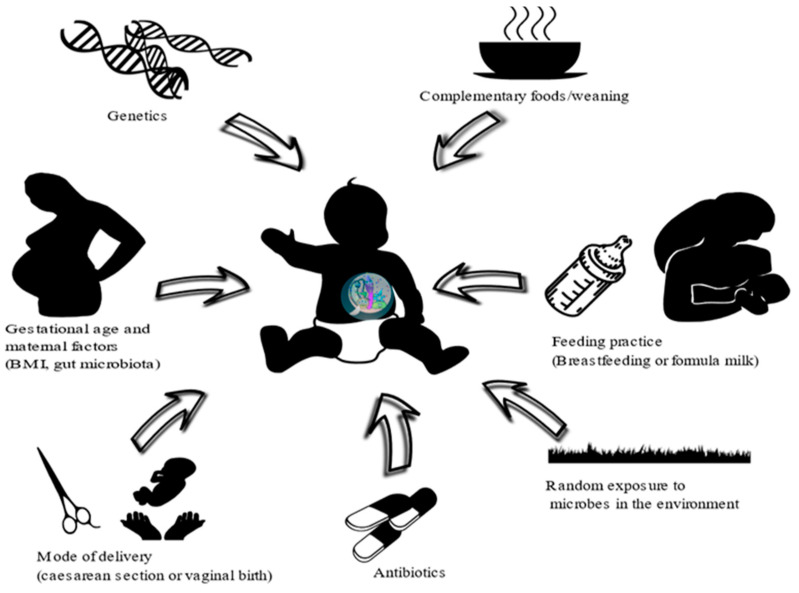
Major factors affecting the microbial composition of the gut in infants.

**Figure 2 nutrients-13-02727-f002:**
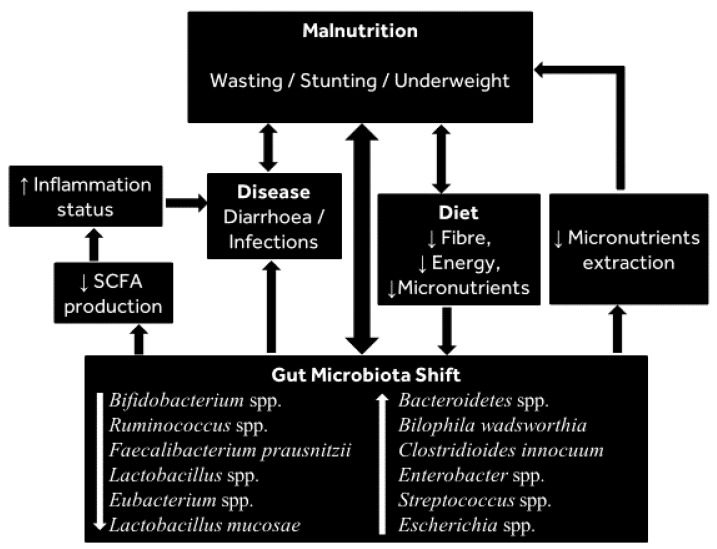
Summary of changes in the gut microbiota associated with childhood malnutrition and key factors linked with such changes. The gut microbiota changes summarise those presented in Table 1, and show reductions in potentially beneficial species and increases in potentially harmful species. Arrows indicate factors that increase or decrease during childhood malnutrition.

## Data Availability

Not applicable.

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
