# Peer review of "Malnutrition and Gut Microbiota in Children"

_nutrients, 2021, doi:10.3390/nu13082727_

Round 1

Reviewer 1 Report

Overall, I thought this was generally well written and interesting. However, there were quite a few factual errors, and I also thought some of the text was overly simplistic/lacking in necessary nuance. I have indicated where edits are necessary in the “Specific comments” section below. I apologise for the length of these comments, but they are meant to be constructive rather than destructive, and I hope that incorporation of them will improve the accuracy of the text.

More broadly, I thought that the review should also mention Environmental Enteric Dysfunction (EED - https://www.ncbi.nlm.nih.gov/pmc/articles/PMC4920693/), and the role this plays in malnutrition as the gut microbiota could plausibly play multiple roles in influencing EED development and progression (e.g., colonisation resistance, immunomodulatory effects, strengthening of the epithelial barrier via SCFA production etc). I therefore recommend that the authors add some text on this topic, perhaps linked to their section on diarrhoeal disease?

Similarly, it is also missing discussion of some key recent articles, showing how microbiota-targeted dietary interventions have the potential to reduce the incidence of malnutrition/stunting (see https://www.ncbi.nlm.nih.gov/pmc/articles/PMC7993600/ and https://pubmed.ncbi.nlm.nih.gov/31296738/ for examples). This seems far more pertinent to this review article than the RUTF-based studies they comment on in some detail in the current version of the text. I therefore recommend that they fully discuss these other dietary intervention studies in the review article too.

Specific comments:

Line 58 – it states “10^12 CFU/mL” here, but the normal range for microbial cell density in the colon is between 10^10 and 10^11 cells per ml.

Lines 76 to 83 – Given the structural differences between HMOs and inulins/oligosaccharides it is often different species of bifidobacteria that are promoted by prebiotics? There are of course huge variations in phenotype between different species, and even strains, of bacteria. HMO-specific species/strains may be particularly important in the infant gut, and will not necessarily be promoted by prebiotics?

Lines 85 to 91 – I thought this section was too unbalanced. There are lots of studies showing little or no effect of probiotics, which are not discussed here. In any event, the term “probiotic” is too broad to assign clinical efficacy. It would be much better to add these caveat points, and perhaps pick out specific probiotic products that have been repeatedly shown to be effective, rather than cherry pick individual studies.

Lines 93 to 95 – are causative roles (in humans, not mice) really confirmed for all of the conditions listed here? I would argue that for at least some of them (mood disorders, obesity, autism) there is not definitive proof yet.

Line 98 – should be “Clostridioides difficile”

Lines 99 to 107 – As is clear from this section, there really is not a lot of consistency between studies regarding which species are linked to weight gain/obesity in children. Indeed, in adults, three recent meta-analyses have concluded there is no taxonomic microbiota signature associated with obesity (http://mbio.asm.org/content/7/4/e01018-16; http://journals.plos.org/plosone/article?id=10.1371/journal.pone.0084689; http://onlinelibrary.wiley.com/doi/10.1016/j.febslet.2014.09.039/full). I would argue that, actually, this really isn’t great evidence for specific members of the gut microbiota being causally associated. This section is therefore an example of one which is selectively descriptive and lacking in nuance and balance.

Line 113 – what is meant by “reduction in SCFA-producing bacteria” here? Most gut anaerobes make SCFAs during fermentation – they can’t all be reduced?

Line 124 – “imbalance” is a little pseudoscientific? What specific features of the microbiota would consistently indicate an “imbalance” that would be always be clinically significant?

Lines 129 to 133 (and Figure 1) – this is a good list but neglects, arguably, one of the greatest influence on the gut microbiota development: random/stochastic exposure to microbes in the environment. I recommend adding this to the text and figure.

Line 144 – “complimentary” should be “complementary”

Lines 153 to 157 – Roseburia and Eubacterium are extremely oxygen intolerant, do they really survive in breast milk and transmit to the infant via that route? How do they get into breast milk in the first place?

Lines 159 to 162 – the use of the word “dominate” is problematic here. Bifids are the most dominant taxa in breast-fed infants, and usually much more so than streps or lactobacilli, and C. difficile, Enterobacter cloacae, Citrobacter spp., Granulicatella adiacens and Bilophila wadsworthia certainly do not “dominate” the microbiota of formula-fed infants. Instead, it is a much more varied community in formula-fed infants, consisting of dominant taxa such as Bacteroides and bifids rather than the listed opportunistic pathogens, which might be more prevalent, but are usually in relatively low proportional abundance.

Figure 1 – “Cesarean” is spelled “caesarean” earlier in the text, so it’s probably best to pick one spelling variant and keep the same throughout (UK English spelling seems to be the predominant choice in other sections of the text?). Also, does it need a capital “M” in “Milk”?

Line 223 – “utilises” should be “utilise”?

Figure 2 – spelling mistake in “Bacteriodetes”

Page 7 – mix of US and UK English spelling of words based on diarrhea/diarrhoea. Suggest standardising one way or the other.

Lines 293 to 294 – perhaps define “severe weight loss”? “Severe” is a subjective term and could be construed differently by individual readers. What proportion of body weight did the mice lose?

Lines 308 to 312 – is Butyrivibrio an established potential pathogen? Have Koch’s postulates been fulfilled for this genus?

Table 1 – Perhaps missing some key studies (e.g. https://pubmed.ncbi.nlm.nih.gov/31296739/)?

Lines 328 to 337 – This text is generally good, but I think it is fair to say that there is some variation between studies with respect to the types of bacteria that are “different” when comparing malnourished to healthy cohorts? The authors may therefore wish to add this caveat somewhere here too, or specifically mention taxa that are most consistently reported as different between studies (see Table 1) rather than/in addition to just mentioning Bilophila wadsworthia and Clostridium innocuum from mouse studies? I think summarising Table 1, giving the most consistent findings, would be a very useful addition.

Line 351 - spelling mistake in “Bacteriodetes”

Lines 351 to 352 – I wondered why bacteriocins and immune stimulation were specifically mentioned here in this way? Other gut bacteria will make bacteriocins (or other antimicrobials) and stimulate the immune system, not just the taxa in the preceding sentence. I suggest re-wording this sentence to make it less speculative, and add more nuance.

Author Response

We thank the reviewer for the comments and suggested corrections. The comments have been addressed below.  

More broadly, I thought that the review should also mention Environmental Enteric Dysfunction (EED - https://www.ncbi.nlm.nih.gov/pmc/articles/PMC4920693/), and the role this plays in malnutrition as the gut microbiota could plausibly play multiple roles in influencing EED development and progression (e.g., colonisation resistance, immunomodulatory effects, strengthening of the epithelial barrier via SCFA production etc). I therefore recommend that the authors add some text on this topic, perhaps linked to their section on diarrhoeal disease?

Many thanks for this important point, this is now included (lines 336-348):

It has been reported that certain subclinical alterations in the gut microbiome can lead to stunting even in the absence of obvious infections such as diarrhoea [121-123]. For instance, poor sanitary conditions whereby there is chronic exposure to environ-mental pathogens resulting in subclinical alteration in the gut microbiota structure and function has been proposed to cause stunting [124, 125]. This has been termed envi-ronmental enteric dysfunction (EED). EED is defined as an acquired subclinical disor-der of the small intestine, marked by villous atrophy and crypt hyperplasia. EED often affects children in resource poor countries [121] and may contribute to stunting. In Gambian infants, faecal neopterin, which has been linked with cell-mediated inflam-mation and is a marker of EED, was negatively associated with weight and height gain [126]. Consistently, in a pilot study conducted in 8 different countries observed a sig-nificant association between faecal neopterin, myeloperoxidase and alpha-1-antitrypsin, and a decrease in length-for-age z-scores (an indicator of stunting in chil-dren) [127].

Similarly, it is also missing discussion of some key recent articles, showing how microbiota-targeted dietary interventions have the potential to reduce the incidence of malnutrition/stunting (see https://www.ncbi.nlm.nih.gov/pmc/articles/PMC7993600/ and https://pubmed.ncbi.nlm.nih.gov/31296738/ for examples). This seems far more pertinent to this review article than the RUTF-based studies they comment on in some detail in the current version of the text. I therefore recommend that they fully discuss these other dietary intervention studies in the review article too.

Added (lines 437-462) and in Table 1:

A recent study comparing the effects of two ‘microbiota-directed complementary food’ (MDCF) and RUTF formulae in Bangladeshi children recovering from SAM confirmed the underdeveloped (immature) nature of the gut microbiome in SAM and post-SAM, MAM children (in comparison with an age-matched control group) [146]. This study further showed that an ‘optimized’ MDCF formula (MDCF-2; containing flour made from chickpea, soyabean, green banana and peanut) can improve a range of growth and other health indicators in MAM children towards the status of healthy controls; this effect was associated with changed abundance of three species most strongly associated with gut microbiota maturation (i.e., increased abundance of F. prausnitzii and Clostridioides spp., and decreased B. longum). A more recent randomized control trial further compared the effect of MDCF-2 and RUSF on growth and health recovery in MAM Bangladeshi children. This research confirmed that MDCF-2 treatment supports growth recovery of MAM children which was linked to corresponding changes in the faecal microbiota abundance of 23 weight discriminatory taxa [145]. The above reports suggest that MDCF treatment can support the immature gut microbiota of malnourished children. Thus, dietary targets that impact the microbiota are worthy of study in order to identify factors that counter malnutrition through microbiota-driven effects. Table 1 summarises the studies (as discussed in this review) on alterations of the gut microbiota of infants and young children during malnutrition.

An important implication of the above studies is that the beneficial changes of dietary interventions (such as RUTF) on the gut microbiota in the treatment of SAM are not sustained once treatment is withdrawn and that further work is required to confirm any beneficial long-term impact of such interventions relating to the gut microbiota of SAM children. However, the recent findings that MDCFs have the potential to ‘repair’ the gut microbiota impairments of childhood malnutrition, leading to improved growth and development, raises the promise of improved future intervention strategies.

Specific comments:

Line 58 – it states “10^12 CFU/mL” here, but the normal range for microbial cell density in the colon is between 10^10 and 10^11 cells per ml.

Corrected (line 58)

Lines 76 to 83 – Given the structural differences between HMOs and inulins/oligosaccharides it is often different species of bifidobacteria that are promoted by prebiotics? There are of course huge variations in phenotype between different species, and even strains, of bacteria. HMO-specific species/strains may be particularly important in the infant gut, and will not necessarily be promoted by prebiotics?

Addressed (lines 94-98):

However, GOS and FOS are structurally different from HMO [38] and so do not entirely replicate the beneficial effects of HMOs on the infant gut microbiota, for example on a species level [34, 39, 40]. Nevertheless, it has been noted that an infant diet supported by prebiotic formula results in fewer infections than a placebo formula, and so non-HMO prebiotics appear to have a positive impact in infants [41].

Lines 85 to 91 – I thought this section was too unbalanced. There are lots of studies showing little or no effect of probiotics, which are not discussed here. In any event, the term “probiotic” is too broad to assign clinical efficacy. It would be much better to add these caveat points, and perhaps pick out specific probiotic products that have been repeatedly shown to be effective, rather than cherry pick individual studies.

Thank you, we have updated the review as suggested (added lines 108-115):

In contrast, numerous studies have failed to find any effect of probiotics on diarrhoea outcomes in children [39, 49, 50]. Indeed, a recent Cochrane review based on large clinical trials with low risk of bias [51], reported little or no effect of probiotics in the reduction of diarrhoea. Such contradictory results are likely to be linked to the use of different probiotics and target populations. Given the contrasting evidence on the use of probiotics in the management of diarrhoea in children, there remains a need for targeted large scale trials to follow on the promising findings of specific probiotics that improve diarrhoea outcomes, especially in children.      

Lines 93 to 95 – are causative roles (in humans, not mice) really confirmed for all of the conditions listed here? I would argue that for at least some of them (mood disorders, obesity, autism) there is not definitive proof yet.

More clarity added (lines 117-122):

There is evidence linking the gut microbial community to a range of diseases and disorders including inflammatory bowel disease (IBD), mood disorders, obesity, autism and psoriatic arthritis [52-57], though further research is needed to ascertain the causal link between the gut microbiota and such diseases. In some of these conditions, interventions that impact the gut microbial community have led to improvement of symptoms, further supporting a role for the microbiota [58-60].

Line 98 – should be “Clostridioides difficile”

Corrected (line 125)

Lines 99 to 107 – As is clear from this section, there really is not a lot of consistency between studies regarding which species are linked to weight gain/obesity in children. Indeed, in adults, three recent meta-analyses have concluded there is no taxonomic microbiota signature associated with obesity (http://mbio.asm.org/content/7/4/e01018-16; http://journals.plos.org/plosone/article?id=10.1371/journal.pone.0084689; http://onlinelibrary.wiley.com/doi/10.1016/j.febslet.2014.09.039/full). I would argue that, actually, this really isn’t great evidence for specific members of the gut microbiota being causally associated. This section is therefore an example of one which is selectively descriptive and lacking in nuance and balance.

This has been updated to consider the studies in a more balanced way

(lines 137-140):

However, caution is required when considering causal links between the gut microbiota and obesity, as recent meta-analyses failed to find variations in the taxonomic microbial compositions of obese and lean adults, suggesting that microbiota differences observed in other studies could be related to factors such as diet [64-66].

Line 113 – what is meant by “reduction in SCFA-producing bacteria” here? Most gut anaerobes make SCFAs during fermentation – they can’t all be reduced?

Addressed (lines 143-147):

For instance, some studies show that infants with eczema have a significant reduction in relative abundance of Bifidobacterium, Blautia, Coprococcus, Eubacterium and Propionibacterium species [67] as well as a reduction in intestinal microbial population diversity [68, 69]; although it is worth noting that in healthy infants, there is low microbiota diversity predominated by bifidobacteria.

Line 124 – “imbalance” is a little pseudoscientific? What specific features of the microbiota would consistently indicate an “imbalance” that would be always be clinically significant?

Addressed (lines 154-159):

………. whilst an altered balance, as characterized by altered abundance of key species (e.g. pathogenic taxa such as Clostridioides difficile), can be associated with negative-health outcomes. However, one drawback of focusing on changes in the microbiota is that it is the difficulty in determining the clinical relevance of such changes and so it remains unclear to what degree the gut microbiota contributes to malnutrition in children (see section 4.3).

Lines 129 to 133 (and Figure 1) – this is a good list but neglects, arguably, one of the greatest influence on the gut microbiota development: random/stochastic exposure to microbes in the environment. I recommend adding this to the text and figure.

This is added now in the text (line 166) and in figure 1:

………., random exposure to microbes in the environment ………………

Line 144 – “complimentary” should be “complementary”

Corrected (line 183)

Lines 153 to 157 – Roseburia and Eubacterium are extremely oxygen intolerant, do they really survive in breast milk and transmit to the infant via that route? How do they get into breast milk in the first place?

This is corrected (lines 192-194):

Many butyrate-producing bacteria such as Roseburia and Eubacterium have recently been demonstrated to grow on complex HMO [84]

Lines 159 to 162 – the use of the word “dominate” is problematic here. Bifids are the most dominant taxa in breast-fed infants, and usually much more so than streps or lactobacilli, and C. difficile, Enterobacter cloacae, Citrobacter spp., Granulicatella adiacens and Bilophila wadsworthia certainly do not “dominate” the microbiota of formula-fed infants. Instead, it is a much more varied community in formula-fed infants, consisting of dominant taxa such as Bacteroides and bifids rather than the listed opportunistic pathogens, which might be more prevalent, but are usually in relatively low proportional abundance.

This has been addressed (lines 199-205):

  1. longum subsp. infantis and Lactobacillus acidophilus have been observed to be higher in abundance in the gut microbiota of breast- as compared to formula-fed infants whereas C. difficile, Enterobacter cloacae, Citrobacter spp., Granulicatella adiacens and Bilophila wadsworthia abundance is higher for formula- compared to breast-fed infants. Additionally, Ruminococcaceae, Bacteroides spp. and Lachnospiraceae of the Clostridia order are of raised abundance in the gut microbiota of infants on complementary feeding [91-94].

Figure 1 – “Cesarean” is spelled “caesarean” earlier in the text, so it’s probably best to pick one spelling variant and keep the same throughout (UK English spelling seems to be the predominant choice in other sections of the text?). Also, does it need a capital “M” in “Milk”?

Corrected (Figure 1)

Line 223 – “utilises” should be “utilise”?

Corrected (line 268)

Figure 2 – spelling mistake in “Bacteriodetes”

Corrected in figure 2

Page 7 – mix of US and UK English spelling of words based on diarrhea/diarrhoea. Suggest standardising one way or the other.

Corrected across text

Lines 293 to 294 – perhaps define “severe weight loss”? “Severe” is a subjective term and could be construed differently by individual readers. What proportion of body weight did the mice lose?

This has been added (lines 355-356)

… caused severe weight loss (about 35% weight loss from the initial body weight), …..

Lines 308 to 312 – is Butyrivibrio an established potential pathogen? Have Koch’s postulates been fulfilled for this genus?

Corrected (lines 374)

Table 1 – Perhaps missing some key studies (e.g. https://pubmed.ncbi.nlm.nih.gov/31296739/)?

This and couple of other relevant studies added to table 1 as recommended

Lines 328 to 337 – This text is generally good, but I think it is fair to say that there is some variation between studies with respect to the types of bacteria that are “different” when comparing malnourished to healthy cohorts? The authors may therefore wish to add this caveat somewhere here too, or specifically mention taxa that are most consistently reported as different between studies (see Table 1) rather than/in addition to just mentioning Bilophila wadsworthia and Clostridium innocuum from mouse studies? I think summarising Table 1, giving the most consistent findings, would be a very useful addition.

This is a good suggestion, now added (lines 397-401):

However, those taxa consistently reported to decrease during malnutrition include Bifidobacterium spp., Lactobacillus spp., Ruminococcus spp. and Faecalibacterium prausnitzii, whilst those consistently reported to increase include Bacteroidetes, Clostridioides innocuum, Sreptococcus spp. and Escherichia spp. (see figure 1 for summary of taxa consistently shifting during malnutrition).

Line 351 - spelling mistake in “Bacteriodetes”

Corrected (line 422)

Lines 351 to 352 – I wondered why bacteriocins and immune stimulation were specifically mentioned here in this way? Other gut bacteria will make bacteriocins (or other antimicrobials) and stimulate the immune system, not just the taxa in the preceding sentence. I suggest re-wording this sentence to make it less speculative, and add more nuance.

This is corrected (lines 423-425):

These changes are predicted to protect against enteropathogens by contributing to the production of bacteriocins and supporting the stimulation of the immune system. Such changes were only partially maintained once RUTF was withdrawn.

Reviewer 2 Report

In the manuscript ID- nutrients-1296916 titled “Malnutrition and gut microbiota in children” by Ishawu Iddrisu and colleagues. The authors have reported that diarrhea, a major contributor to malnutrition, is induced by pathogenic elements of the gut microbiota. Diarrhea leads to malabsorption of essential nutrients and reduced energy availability resulting in weight loss, which can lead to malnutrition. Alterations in the gut microbiota of severe acute malnourished (SAM) children include increased Proteobacteria and decreased Bacteroides levels. Additionally, the gut microbiota of SAM children exhibits lower relative diversity compared to healthy children. Thus, the data indicate a link between gut microbiota and malnutrition in children suggesting that treatment of childhood malnutrition should include measures that support healthy gut microbiota. This could be of particular relevance in sub-Saharan Africa and Asia where the prevalence of malnutrition remains a major threat to the lives of millions. I have few concerns regarding the present manuscript.

-I enjoy the reading of the entire document, the idea is really good, however, some changes are required

-The microbiota in health and disease need emphasizes more in nutritional aspects and the relationship with other diseases and connection, brain-gut-axis

-More detailed information about the delivery mode and breastfeeding is required

-Other treatments for malnutrition need to be add to the document

-Further directions are required in this manuscript

Author Response

We thank the reviewer for the suggestions and comments, we have founds these to be very useful. Please see below and in the manuscript our changes.

The microbiota in health and disease need emphasizes more in nutritional aspects and the relationship with other diseases and connection, brain-gut-axis

We thank the reviewer for this comment. We have included more information on the microbiota involvement in the gut brain-axis (lines 66-76):

The gut microbiota has also been observed to an play important role in the absorption, storage and expenditure of energy from the diet [17-19] as well as the synthesis of vitamins K and B12 [20, 21]. Energy intake and expenditure is predominantly coordinated by the brain. The gastrointestinal tract (GI), the first site of contact with ingested food, sends important signals (through neuroendocrine and neuroimmune systems) to the brain regarding the composition and size of incoming food [22]. Furthermore, the gut microbiota is involved in the production of metabolites including SCFAs, secondary bile acids tryptophan and even neurotransmitters; these play an important role in gut barrier function and subsequently in gut-brain signaling [23, 24]. The gut-brain axis is important for maintaining energy balance and controlling energy expenditure, and has also been observed to impact cognitive function [25].

More detailed information about the delivery mode and breastfeeding is required

Details of mode of delivery have now been added (lines 173-180):

In a recent longitudinal study, delayed microbiota colonization and significant taxonomic differences were observed for Caesarean deliveries compared to vaginal deliveries in infants during their first year, although such differences were lessened 5 months after birth [85]. Also, a recent clinical study on the effect of mode of delivery on gut microbiota composition [86] showed an enrichment of Bifidobacterium spp., and a reduction in Klebsiella and Enterococcus spp. in the gut microbiota of vaginally-delivered infants compared to Caesarean section born infants. This has also been shown in a more recent study [87] where Caesarean section born infants were found to be less likely to have B. thetaiotaomicron or B. fragilis in their gut microbiota.

Details of breastfeeding has now been added (lines 199-211):

  1. longum subsp. infantis and Lactobacillus acidophilus have been observed to be higher in abundance in the gut microbiota of breast- as compared to formula-fed infants whereas C. difficile, Enterobacter cloacae, Citrobacter spp., Granulicatella adiacens and Bilophila wadsworthia abundance is higher for formula- compared to breast-fed infants. Additionally, Ruminococcaceae, Bacteroides spp. and Lachnospiraceae of the Clostridia order are of raised abundance in the gut microbiota of infants on complementary feeding [91-94]. A recent clinical study observed an enrichment of Bifidobacterium and Bacteroides spp., and a lower abundance of Streptococcus and Enterococcus spp., in the gut microbiota of breast-fed compared to formula-fed infants [95]. In addition, as indicated above, breast milk has been observed to contain bifidobacteria and to promote a bifidogenic effect [83]. Thus, breastfeeding has a major impact on the composition of the infant gut microbiota, driving a predominance of bifidobacteria, during the first year of life [96].

Other treatments for malnutrition need to be add to the document

This is a good point and has now been added (lines 411-415) however the emphasis is on the treatment recommended by WHO:

In line with the recommendations of WHO inpatient treatment of SAM, a milk-based formula (F75) is used for the stabilization phase of treatment before energy/micronutrient dense RUTF treatment is initiated. For community-based treatment of moderate acute malnutrition (MAM), Ready-to-Use Supplementary Foods (RUSF) are used.

Future directions are required in this manuscript

These has now been included (lines 480-486):

Future Direction: Targeting the gut microbiota, such as through the use of prebiotics, as part of the treatment of childhood malnutrition could present an effective, affordable and sustainable strategy in the treatment and/or prevention of SAM. Fortifying the microbiota in this way could enhance energy salvaging and may protect against diarrhoea. However, clinical evidence that can be translated into clinical practice is lacking. Therefore, studies on the impact of prebiotics on the gut microbiota and its influence on nutrition outcomes in children suffering from malnutrition are needed.

Reviewer 3 Report

These manuscript are well written and of scientific sound.

On line 285: Where the authors state that "Therefore, it is possible that 
pre- and probiotic interventions may help to support the microbiota to combat diarrhoea" I suggest that the speakers refer to the work of Suez (Jotham Suez et al) on the impact of probiotics and antibiotic resistance especially when talking about undernaurished children who might be on antibiotics where probiotics may actually increase the antibiotic resistance gene resevours (Probiotics impact the antibiotic resistance gene reservoir along the human GI tract in a person-specific and antibiotic-dependent manner | Nature Microbiology).

Author Response

Many thanks for the thoughtful comments on our manuscript. Please see how we have incorporated your comments below and on the attached.

On line 285: Where the authors state that "Therefore, it is possible that 
pre- and probiotic interventions may help to support the microbiota to combat diarrhoea" I suggest that the speakers refer to the work of Suez (Jotham Suez et al) on the impact of probiotics and antibiotic resistance especially when talking about undernaurished children who might be on antibiotics where probiotics may actually increase the antibiotic resistance gene resevours (Probiotics impact the antibiotic resistance gene reservoir along the human GI tract in a person-specific and antibiotic-dependent manner | Nature Microbiology).

We thank the reviewer for this comment. We had previously looked at the paper by Suez et al, however we think it has limitations that undermine the interpretation of the paper. For example there were just 8 volunteers consuming antibiotics and the endpoint was microbial diversity and recovery, rather than clinical outcomes. There are in fact, many studies indicating a positive impact of supporting the microbiota whilst antibiotics are consumed. We do agree that it is a relevant point that antibiotics use during treatment of SAM needs to be considered and we have now mentioned the effect of prebiotics alongside antibiotics use on the gut microbiota (lines 331-335):

The use of prebiotics may be of particular benefit to the management of SAM where children receive antibiotics as part of the treatment process. Prebiotics have been shown to reduce the adverse effects of antibiotics on the gut microbiota [120], however further research is needed with regards to their impact in the treatment of malnutrition.

Round 2

Reviewer 1 Report

I thank the authors for making the requested changes to the manuscript.

I have no further comments, except to point out the spelling mistake in "Sreptococcus" on line 396, which I presume can be corrected at the proofing stage.

Reviewer 2 Report

Thank you to the authors for taking into account my previous comments. No further comments are required, thanks again